# Advancing Cancer Treatment: A Review of Immune Checkpoint Inhibitors and Combination Strategies

**DOI:** 10.3390/cancers17091408

**Published:** 2025-04-23

**Authors:** Valencia Mc Neil, Seung Won Lee

**Affiliations:** 1Department of Precision Medicine, Sungkyunkwan University School of Medicine, Suwon 16419, Republic of Korea; hy.neilll@skku.edu; 2Department of Artificial Intelligence, Sungkyunkwan University, Suwon 16419, Republic of Korea; 3Department of Metabiohealth, Sungkyunkwan University, Suwon 16419, Republic of Korea; 4Personalized Cancer Immunotherapy Research Center, Sungkyunkwan University School of Medicine, Suwon 16419, Republic of Korea

**Keywords:** oncolytic virus, cancer vaccine, glutamine metabolism, immune checkpoint inhibitor synergy, microbiota-metabolite-immune, lung cancer, hepatocellular carcinoma melanoma, breast cancer

## Abstract

Cancer treatment has advanced greatly over the past decades of experiments and trials with immune checkpoint inhibitors, which help the immune system recognize and attack cancer cells. Chemotherapy is a standard and common treatment for cancer; however, it can also damage healthy cells. Immunotherapy has shown promising efficacy in lung, liver, melanoma, and triple-negative breast cancers (TNBCs). However, its effectiveness is limited because some patients are resistant to this treatment. This review explores the latest combination strategies to enhance the response to treatment, including radiotherapy, metabolic reprogramming, microbiome modulation, and dual checkpoint blockade. In addition, biomarkers and novel immune checkpoints are being explored to optimize patient outcomes. This review aims to improve ICI-based therapies and guide future cancer research by identifying effective combination approaches and understanding resistance mechanisms. The findings are not only essential for clinicians to develop more personalized cancer treatments but also to offer new hope to patients with hard-to-treat cancers.

## 1. Introduction

A groundbreaking milestone in oncology has been the recognition and targeted elimination of malignant cells through cancer immunotherapy, which harnesses the body’s immune system to attack cancer. Several therapies have been utilized in cancer treatment, and among the most promising advances in this field of cancer therapies are immune checkpoint inhibitors (ICIs), which block inhibitory pathways that limit immune responses and enhance antitumor immunity. ICIs—such as programmed death-1 (PD-1), programmed death-ligand 1 (PD-L1), and cytotoxic T-lymphocyte-associated protein 4 (CTLA-4)—have shown remarkable efficacy in treating malignancies, including hepatocellular carcinoma (HCC) and melanoma [1]. Despite their robust efficacy in cancer treatment and clinical benefits, many patients still exhibit either primary or acquired resistance to ICIs, which has necessitated the development of combination strategies to improve treatment outcome [2].

Combining ICIs with radiotherapy, targeted therapies, or modulation of the gut microbiota has been shown to enhance therapeutic efficacy by overcoming resistance mechanisms and modifying the tumor microenvironment (TME). For example, *Bifidobacterium* enhances the antitumor efficacy of PD-L1 blockade by promoting dendritic cell (DC) maturation and increasing CD8+ T-cell priming within tumor microenvironment (TME) [3,4]. Jhawar et al. [5] demonstrated that oncolytic virus (OV) therapy and radiotherapy (RT) suppress tumor growth by converting immunologically “cold” tumors through CD8+ T cell- and IL-1α-dependent mechanisms, resulting in increased PD-1/PD-L1 expression. The triple combination of OV, RT, and PD-1 inhibitors not only delays tumor progression but also extends survival. In a clinical example, a PD-1-refractory patient with cutaneous squamous cell carcinoma experienced unexpected long-term tumor control and survived for more than 44 months following triple therapy.

Furthermore, results from a skin cancer mouse model revealed that OV, RT, and ICI therapies enhanced CD8+ T-cell infiltration and IL-1α expression [5]. Radiotherapy has also been employed across various cancer types to synergize with ICIs by inducing immunogenic cell death and enhancing tumor antigen presentation [6]. Targeted therapies, such as BRAF/MEK inhibitors in melanoma and TKIs in HCC, modulate immune responses, resulting in tumor susceptibility to checkpoint blockade [7]. Preclinical studies support the combination of BRAF or BRAF/MEK inhibitors with immune checkpoint blockade (ICB) in melanoma treatment. Inhibition of PD-1 or PD-L1 enhances the efficacy of both BRAF inhibitor monotherapy and dual BRAF/MEK inhibition in BRAFV600 melanoma mouse models. This effect is CD8+ T-cell-dependent and associated with increased numbers and improved functionality of tumor-infiltrating lymphocytes. Moreover, resistance to BRAF inhibitors has been associated with increased PD-L1 expression in tumor cells, resulting in PD-1 blockade, a strategy to restore or boost the effectiveness of BRAF inhibition [7,8,9].

### 1.1. Materials and Methods

The literature search included publications from 2015 to 2025, with older studies retained only for comparison with recent cancer treatment advancements. Articles were retrieved from PubMed, Web of Science, and Google Scholar. As this is not a systematic review, it does not adhere to the PRISMA guidelines. To ensure a comprehensive review and information, various keywords were used, including “immune checkpoint inhibitors (ICIs)”, “PD-1/PD-L1 blockade”, “CTLA-4 inhibitors”, “combination therapy”, “tumor microenvironment”, “ICI resistance”, “radiotherapy”, “metabolic reprogramming”, “gut microbiome”, “oncolytic viruses”, “novel immune checkpoints (LAG-3, TIM-3, TIGIT) “, “case study” and “real-world data.” Eligible articles included systematic reviews, clinical trials, meta-analyses, and multinational, multicenter, retrospective studies, as well as guidelines related to ICI therapy in NSCLC, HCC, melanoma, and TNBC. The primary aim of this study was to evaluate the efficacy of ICIs in cancer treatment, provide real-world evidence, examine combination strategies, and analyze biomarker-driven approaches to immunotherapy. The most recent and significant findings are summarized in tables and figures to highlight key trends and clinical outcomes.

### 1.2. Immunotherapy in Oncology

Cancer remains the second leading cause of death globally, following cardiovascular and cerebrovascular diseases, with incidence steadily increasing. In 2022, nearly 20 million new cancer cases, including non-melanoma skin cancers (NMSCs), were reported, along with 9.7 million cancer-related deaths worldwide. Currently, approximately one in five individuals is expected to develop cancer during their lifetime, with mortality rates estimated at one in nine for men and one in twelve for women [10].

Surgery, radiotherapy, and chemotherapy are traditional primary approaches to cancer treatment. However, with the success of immunotherapy—both as a standalone treatment and in combination with other modalities—it has been recognized as the fourth pillar of cancer therapy [11]. Immunotherapy leverages the body’s immune system, unlike conventional therapies that directly target tumor cells, to identify and eliminate cancer cells, providing a more natural and adaptive approach to controlling disease progression than conventional therapies. Despite its promise, cancer cells can activate alternative signaling pathways to escape targeted therapies. In addition, the complexity of the TME and interactions between cancer class and stromal components diminish the effectiveness of single-drug treatments [12].

In contrast to traditional therapies, immunotherapy works by modulating the TME through cytokines, chemokines, and immune cells, enhancing immune responses and promoting sustained tumor control while reducing recurrence. This treatment paradigm shift has significantly expanded the treatment options for improving outcomes in many patients with cancer [13]. Over the past decade, FDA-approved ICIs—such as nivolumab, cemiplimab, atezolizumab, ipilimumab, and durvalumab, have emerged as transformative advances in oncology, revolutionizing treatment across multiple malignancies (Figure 1). Furthermore, due to variable patient responses, combination treatments are significantly increasing, as they enhance effectiveness and broaden applicability [14].

Table 1 summarizes the outcomes of multiple clinical and real-world studies evaluating ICI therapies. Notably, immunotherapy consistently demonstrates higher objective response rates (ORR), disease control rates (DCR), and longer PFS and OS. These underscore ICI strategies in clinical settings, particularly in patients with advanced or treatment-resistant cancers, emphasizing the need for tailored approaches that consider patient-specific factors such as tumor biology, immune landscape, and performance status.

## 2. Role of Immune Checkpoint Inhibitors in Selected Cancers

The development of ICI, such as CTLA-4 and PD-1/PD-L1 inhibitors, has transformed cancer therapy by demonstrating remarkable antitumor efficacy. This has led to its widespread clinical application as a monotherapy or in combination with other treatments. ICI has become a standard treatment option for various solid tumors, including those with microsatellite instability-high (MSI-H) status, since the approval of ipilimumab, a CTLA-4-targeting monoclonal antibody, in 2011 for advanced metastatic melanoma. Several immune checkpoints are currently being investigated for their therapeutic potential, progressing from preclinical research to clinical trials [22]. Another breakthrough in ICI therapy is relatlimab, an FDA-approved LAG-3 inhibitor [23] used in combination with nivolumab to treat unresectable or metastatic melanomas. Clinicians aim to identify novel immune checkpoints that optimize combination therapies with complementary antitumor mechanisms to broaden the clinical applications and therapeutic efficacy of ICIs [24].

Numerous data and evidence suggest that immunotherapy significantly improves PFS and OS in patients with cancer compared with conventional treatments. Table 2 summarizes the ICI response rates, PFS, OS, and DCR for the leading cancer types across the globe, such as NSCLC, HCC, melanoma, and TNBC. ICIs have demonstrated substantial efficacy in managing advanced metastatic and highly immunogenic tumors, such as melanoma and Merkel cell carcinoma. Various cancer treatments utilizing immune checkpoint blockade are currently under investigation in clinical trials, including CTLA-4 inhibitors and anti-PD-1 antibodies, which have shown enhanced antitumor activity [25,26].

### 2.1. Non-Small Cell Lung Cancer (NSCLC)

NSCLC is a major global health concern due to its high prevalence and mortality rate, accounting for approximately 85% of all lung cancer cases and making it the most common form of lung cancer. In 2022, nearly 20 million new cancer cases were reported worldwide, including non-melanoma skin cancers (NMSCs), with approximately 9.7 million cancer-related deaths. Demographic and statistical projections indicate that the incidence of new cancer cases could rise to 35 million by 2050 [10].

A systematic review was conducted by Xiao et al. [27], who retrieved randomized controlled trials (RCTs) from PubMed, Embase, and the Cochrane Library that compared neoadjuvant or adjuvant immune checkpoint inhibitors (Neo/Adj ICI) plus chemotherapy versus chemotherapy alone in resectable NSCLC. Among 3926 patients, neoadjuvant ICI significantly increased pathological complete response (pCR) by fivefold (OR 5.35, 95% CI) compared to chemotherapy alone. The risk was reduced by 34% (HR 0.66) for both neo- and adjuvant ICI use compared to monotherapy. Two-year OS improved with the inclusion of either neo- or adjuvant ICI, with median OS (HR 0.76). Moreover, neoadjuvant ICI enhanced major pathological response (MPR) nearly fourfold (OR 3.91). However, patients with non-squamous NSCLC, ever smokers, and tumors with PD-L1 expression ≥ 1% experienced the most benefit from ICI therapy. This preliminary finding shows that the integration of Neo/Adj ICI into treatment regimens for resectable NSCLC improves the recurrence rate with a potential OS advantage.

A retrospective cohort study conducted by Zhang et al. [19] at Peking University Cancer Hospital in 110 elderly patients diagnosed with NSCLC and treated with either chemotherapy alone, a combination of ICIs (pembrolizumab, atezolizumab, camrelizumab, and infliximab) and chemotherapy (ICI plus pemetrexed, gemcitabine, albumin-bound paclitaxel/platinum-doublet treatment), ICI monotherapy, or ICIs combined with other therapies such as anti-angiogenic agents (bevacizumab, endostar) or novel immune checkpoint inhibitors revealed that the ICI combination group achieved the highest DCR at 75%. PFS and OS showed no significant differences between the ICI + chemo and ICIs groups (median PFS 5.3 months vs. 5.5 months), patients receiving chemotherapy alone had a shorter median PFS of 3.9 months. Median OS for ICI monotherapy and chemotherapy was 10.9 months, 10.7 months compared to 20.3 months for ICI combination therapy. In subgroup analysis, patients with PD-L1 expression ≥1% exhibited a trend toward improved OS in the ICIs group than in the ICI + chemotherapy group (22.4 vs. 10.7 months), although the difference was not statistically significant. In terms of tolerability, ICIs had a lower treatment discontinuation rate owing to adverse effects than the ICI + chemotherapy group. Thus, ICI monotherapy or ICIs combined with anti-angiogenic agents may be preferable to chemotherapy in elderly PD-L1-positive patients.

A study by Miao et al. [28] involving 351 NSCLC patients who received ICI therapy at the Lung Cancer Center of Peking Union Medical College Hospital between 2018 and 2021 revealed that the median PFS for patients receiving ICI-based therapies was 9.5 months, and an overall response rate (ORR) of 47.3%. Meanwhile, the median PFS and ORR varied across different ICI. In one instance, pembrolizumab had a median PFS of 9.6 months with an ORR of 45.0%, and nivolumab had a median PFS of 9.2 months with an ORR of 50%. Camrelizumab demonstrated the longest median PFS at 10.4 months, with an ORR of 47.6%, whereas tislelizumab had a median PFS of 10.3 months and achieved the highest ORR of 54.2%. The majority of patients in this study were over 60 years of age, in contrast to the findings of another study, which showed that the efficacy and patient tolerance of ICIs were not significantly correlated with age [41]. However, Miao et al. [28] demonstrated that patients aged 60–74 had longer PFS. Notably, male patients showed benefits from ICI therapy regardless of age. More than 75% of the patients included in the present study were aged > 60 years. The efficacy of ICIs, as well as the patient tolerance of ICIs, was not significantly correlated with age according to previous studies. However, Miao et al. [28] indicated that the 60–74-year-old population exhibited a longer PFS. Meanwhile, males tend to benefit from ICI therapy regardless of age.

Given these studies, integrating ICI in treatment regimens for NSCLC whether neoadjuvant, adjuvant, or advanced-stage can significantly improve PFS and OS outcome, particularly in patients with PD-L1 ≥ 1%, non-squamous histology, and those aged 60–74. May it be ICI monotherapy or combinations with anti-angiogenic agents, both are tolerated in elderly patients. Clinically, this support tailoring therapy based on patients’ biomarkers status, age and treatment goals, helping guide to personalized and effective NSCLC management.

### 2.2. Hepatocellular Carcinoma

Hepatocellular carcinoma (HCC) is the fifth most prevalent cancer globally and a significant cause of cancer-related deaths, particularly in South Korea. HCC is associated with chronic liver inflammation, fibrosis, and complications that disrupt normal liver function [42,43]. Early stage treatment through early detection is essential for effective HCC management and can yield positive outcomes in recurrence-free and overall survival rates [44]. Liver transplantation, surgical resection, and local ablation are key approaches for managing early-stage HCC [43,45].

A multinational, multicenter, retrospective study by Lee et al. [29] analyzed data from 22 centers across five Asia-Pacific countries (APAC) (South Korea, Hong Kong, Taiwan, Thailand, and Singapore), focusing on 1,141 HCC patients who were treated with first-line atezolizumab plus bevacizumab. The median progression-free survival (PFS) and overall survival (OS) for all patients were 2.9 months and 8.0 months, respectively. Sorafenib and lenvatinib are the most commonly used therapies. In second-line treatments, lenvatinib provided superior outcomes over sorafenib, with PFS of 4.0 vs. 2.3 months and OS of 8.0 vs. 6.3 months, respectively. Patients receiving combined tyrosine kinase inhibitors (TKI) and ICI therapy experienced a PFS of 5.4 months and OS of 12.6 months. Furthermore, patients with a lower tumor burden who were treated with lenvatinib or TKI + ICI demonstrated a longer second-line PFS.

Jang et al. [30] examined 101 patients diagnosed with combined hepatocellular-cholangiocarcinoma (cHCC-CCA) treated with ICIs from 2015 to 2021. Within a median follow-up period of 20.1 months, median PFS and OS were 3.5 and 8.3 months, respectively, with an objective response rate (ORR) of 20%. Frequently administered ICI drugs included nivolumab, pembrolizumab, and atezolizumab plus bevacizumab, which have demonstrated anticancer efficacy in the treatment of cHCC-CCA.

Another study conducted by Kim et al. [31] at Asan Medical Center in Seoul analyzed 254 patients with cHCC-CCA with metastatic disease who received systemic chemotherapy for unresectable disease between 1999 and 2015. The ORR was 14.1%, with median PFS and OS at 3.8 and 10.6 months, respectively, for those treated with sorafenib or cytotoxic chemotherapy. Larger intrahepatic tumor burden, higher serum bilirubin levels, and the use of non-platinum chemotherapy are factors associated with poor survival.

A study by Miura et al. [32] at Hiroshima University Hospital from 2020 to 2023 evaluated 16 patients who received atezolizumab–bevacizumab (Atezo + Bev) as first-line therapy and durvalumab–tremelimumab (Dur + Tre) as second-line therapy for unresectable hepatocellular carcinoma (uHCC). The DCR and ORR for Atezo + Bev were 87.5% and 58.3%, respectively, compared to 62.5% and 0% for Dur + Tre. However, the use of TKIs following Atezo + Bev showed a higher response rate in uHCC.

Combination therapy ilvolving ICI has demonstrated strong therapeutic potential, such as transarterial chemoembolization (TACE), ablation, and TKIs therapy. Enhanced efficacy of ICI treatments is observed in TACE and ablation therapy, as they potentially modify the immune environment in cases of disease progression [46,47].

These studies reveals ICI incorporation with HCC treatment including first-line ICI plus bevacizumab or second-line TKI–ICI combinations can improve PFS and OS particularly in patient with lower tumor burden. DC is also improved in cHCC-CCA when ICI-based therapies is followed by TKI treatment. Findings highlight the potential of combining ICIs with targeted or locoregional therapies, supporting more personalized treatment approaches based on tumor burden, histology, and prior therapy responses.

### 2.3. Melanoma

Skin cancer (melanoma of skin and non-melanoma skin cancer) is a dermatological condition with increasing incidence and mortality rates yearly. According to GLOBOCAN 2022, melanoma ranks as the 17th most common cancer globally, with an estimated 331,722 new cases and approximately 58,667 related deaths, affecting both sexes and all ages. The occurrence of skin cancer differs across regions, with the highest incidence reported in Europe (44.1%) and North America (34.0), while much lower incidences were reported in Africa (2.3%) and Asia (7.5%) [48]. Further studies have focused on improving treatment, particularly by exploring new immunotherapy approaches and combination strategies to overcome resistance to current treatments. Although immunotherapy offers promise for treating melanoma, its efficacy is limited due to resistance development that is frequently seen in males. Sex-differences likely influenced by androgen receptor (AR) signaling, affecting immunotherapy outcomes in solid tumors. In a study by Di Donato et al. [49], found depleting AR enhances the effectiveness of ICI particularly atezolizumab. However, anti-PD-1 and anti-CTLA-4 were only effective in one melanoma cell line, suggesting variable responses. Further studies is needed to elucidate AR’s role in tumor immunity and propose combining AR modulation with ICIs as a potential therapeutic strategy.

Namikawa et al. [33] evaluated BRAF/MEK inhibitors (BRAF/MEKi) in 336 patients with advanced BRAF V600-mutant melanoma. Patients were treated with either BRAF/MEKi (dabrafenib + trametinib or encorafenib + binimetinib), anti-PD-1 monotherapy (nivolumab or pembrolizumab), or combination PD-1/CTLA-4 therapy (nivolumab + ipilimumab) as first-line treatment. BRAF/MEKi demonstrated a significantly higher objective response rate (ORR, 69%) and longer progression-free survival (PFS, 14.7 months) compared to anti-PD-1 (ORR: 27%, PFS: 5.4 months) and PD-1/CTLA-4 (ORR: 28%, PFS: 5.8 months). However, overall survival (OS) across the three strategies remained comparable (BRAF/MEKi: 34.6 months vs. anti-PD-1: 37.0 months). Propensity score matching indicated a trend toward longer PFS with BRAF/MEKi and equivalent OS with PD-1/CTLA-4 therapy. Notably, survival outcomes were poorer in patients receiving BRAF/MEKi followed by PD-1/CTLA-4 as a second-line therapy compared to those treated initially with PD-1/CTLA-4 followed by BRAF/MEKi. This suggests that BRAF/MEKi may impair the efficacy of later immunotherapy, underscoring the importance of treatment sequencing. Although BRAF/MEK inhibitors demonstrate longer PFS and similar OS, this discrepancy may be attributed to the development of resistance mechanisms and disease progression after the initial response, which limits the effectiveness of subsequent immunotherapy particularly PD-1/CTLA-4. Conversely, ICIs may take longer to show clinical benefit but can induce more durable responses through sustained immune-mediated tumor control. Interestingly, the potential efficacy of ICIs in Asian patients with BRAF V600-mutant melanoma may be limited by lower TMB, which is associated with reduced UV exposure and Fitzpatrick skin types III–IV, commonly observed in the Japanese population [50]. Additionally, subsequent treatment lines and patient-specific factors such as higher toxicity burden (75% experiencing grade ≥ 3 adverse events), poor Eastern Cooperative Oncology Group performance status (ECOG PS 2–4), advanced disease stage (M1c/M1d), and elevated lactate dehydrogenase (LDH) levels—can influence overall survival outcomes.

Meanwhile, Namikawa et al. [33] highlights the clinical efficacy of BRAF/MEKi in terms of initial response and PFS, it also underscores crucial limitation such as the development of resistance mechanisms and disease progression diminishing the efficacy of immunotherapy. To better understand the biological basis of such resistance, recent investigations have explored the role of androgen receptor (AR) signaling in modulating response to BRAF/MEKi. In a neoadjuvant study involving 51 female patients with melanoma showed higher major pathological response (MPR: 66% vs. 14%) and improved 2-year recurrence-free survival (RFS: 64% vs. 32%) than males. These findings supported by a recent cohort study [51,52] (n = 664), where females demonstrated superior PFS and OS than males. Preclinical models also revealed in male mice that anti-tumor responses diminished to BRAF/MEKi, associated with elevated androgen receptor (AR) expression in tumors, which promotes melanoma cell proliferation, particularly in testosterone-rich environments. Pharmacological AR inhibition (e.g., with enzalutamide) restores sensitivity to BRAF/MEKi in both sexes thus enhances treatment efficacy, and providing a compelling rationale for combination therapy approaches in melanoma.

Building upon the understanding of resistance mechanisms and the potential benefit of AR signaling inhibition in enhancing BRAF/MEKi efficacy, attention has also turned toward optimizing immunotherapy strategies in clinical settings. Zaemes et al. [34] conducted a real-world study using data from a multi-site immuno-oncology registry, including 290 patients with advanced melanoma (AM) and NSCLC. The study found that AM patients who received first-line ICI therapy (anti-CTLA-4, anti-PD-(L)1, or a combination) had a longer treatment-free survival (TFS) of 10.58 months, compared to 8.43 months in those treated with second-line therapy or later treatment. Patients receiving anti-CTLA-4, anti-PD-(L)1, and combination ICI therapies had TFS values of 8.13, 8.56, and 15.04 months, respectively, while those receiving earlier treatment/first-line therapy were notably higher with combination therapy than with AM.

In another retrospective study by Lee et al. [35] on 71 patients, 16 had not previously received pembrolizumab and 55 had prior pembrolizumab treatment diagnosed with histologically confirmed malignant melanoma who underwent chemotherapy at Kyungpook National University Chilgok Hospital in Daegu, Korea, patients received either dacarbazine monotherapy or a CVD regimen (cisplatin, vinblastine, and dacarbazine) for unresectable stage IIIC and IIID shows a median PFS 3.9 months who were treated with pembrolizumab compared to 2.3 months in not previously received pembrolizumab treatment showing 1.6 months difference in PFS. In addition, the median OS was 19.0 months in patients receiving pembrolizumab compared with 6.8 months who had not previously received pembrolizumab treatment. Univariate and multivariate analyses confirmed that pembrolizumab use was associated with better PFS and was the only significant predictor of OS. The mean duration of response (DOR) was 8.11 months vs. 4.64 months, indicating a longer duration in patients who had pembrolizumab exposure. Furthermore, patients with metastatic melanoma who received dacarbazine following pembrolizumab treatment exhibited significantly better PFS and OS than those treated with dacarbazine alone pointing that dacarbazine may still be a valid option incase ICI treatment failed.

Lastly, a retrospective, multicenter, national real-world study by Nardin et al. [36] in 85 patients from 12 centers who had been diagnosed with advanced (unresectable stage III or IV) melanoma that had previously responded to an initial ICI treatment before undergoing ICI rechallenge demonstrated an ORR of 54%, with 30 patients achieving complete response (CR), 16 patients showing partial response (PR), 18 patients having stable disease (SD), and 21 patients experiencing disease progression (PD). Second-line ICI treatments include pembrolizumab, nivolumab, ipilimumab, and ipilimumab plus nivolumab. In addition, PFS was 21 months, whereas the median OS was not reached. However, the 1 and 2 years PFS were 58% and 47%, respectively, while the OS rates were 78% and 71%, respectively. As revealed in this study, responses to ICI rechallenge were associated with better PFS and OS rates than those of non-responders. Furthermore, patients treated with corticosteroids before ICI rechallenge exhibited a poorer OS. Therefore, in patients with advanced melanoma who had previously achieved disease control, the response rate (54%) and disease control rate (75%) were high, suggesting that ICI rechallenge is an effective therapeutic strategy for patients with previously controlled disease.

These studies highlights the integration of ICI and targeted therapies into melanoma treatment, especially first-line BRAF/MEKi or PD-1/CTLA-4 combinations that improve PFS, though overall survival remains comparable. In addition, androgen receptor signaling in resistance suggests AR blockade (e.g., enzalutamide) may enhance BRAF/MEKi efficacy. Improved outcome is seen to patient from ICI rechallenge, combination strategies, and prior ICI exposure (e.g., pembrolizumab).

### 2.4. Triple-Negative Breast Cancer (TNBC)

Triple-Negative Breast Cancer (TNBC) is a highly aggressive form of breast cancer accounting for 12–15% of breast cancer cases in the United States, with a higher incidence in females (23.8%) and both sexes (11.5%) [53]. In the US alone, 297,540 estimated new cases and 43, 170 deaths in 2023 [54]. TNBC is characterized by higher PD-L1 expression levels, with a median PD-L1 combined positive score (CPS) of 7.5, and 50% of cases exhibit a CPS of ≥10 compared to hormone receptor-positive (HR+) breast cancer. TNBC is a high-grade invasive carcinoma of no special type (NST) affecting half of all breast cancer cases among women [55]. In addition, TNBC is associated with elevated levels of tumor-infiltrating lymphocytes (TILs) and tumor mutational burden (TMB), which are associated with a greater response to immunotherapy [56,57].

The phase III KEYNOTE-119 trial evaluated pembrolizumab monotherapy against chemotherapy options (capecitabine, gemcitabine, eribulin, or vinorelbine) by Cortés et al. [58] in 622 patients receiving second- or third-line treatment, revealing no significant differences in PFS and OS between the two groups. The ORR to pembrolizumab increases with higher PD-L1 combined positive scores (CPS), with the greatest benefit observed in patients with CPS ≥ 10. In addition to this study, a phase II trial in 13 breast cancer patients treated with pembrolizumab monotherapy across various solid tumors with microsatellite instability-high (MSI-H) or mismatch repair-deficient (MMR) disease reported a median ORR of 30.8% and a median PFS of 3.5 months [59,60].

Yang et al. [46] conducted a meta-analysis [37] to evaluate the therapeutic efficacy of combining ICIs with chemotherapy (CT) in metastatic TNBC across five English databases (PubMed, Web of Science, CENTRAL, Scopus, and Embase) and four Chinese databases (CBM, CNKI, VIP, and Wanfang), along with oncological conference proceedings comparing ICI plus CT versus CT alone in metastatic TNBC. Five randomized controlled trials (RCTs) with 3000 patients were included in this study. The findings demonstrated that the ICI plus CT regimen significantly improved PFS in both the ITT population (HR 0.80, 95% CI) and PD-L1-positive patients (HR 0.70, 95% CI) compared with chemotherapy alone. Moreover, enhanced OS was observed in the ITT (HR 0.89, 95% CI) and PD-L1-positive (HR 0.80, 95% CI) groups. Furthermore, the study showed an increase in PFS with higher PD-L1 expression, indicating a correlation between PD-L1 enrichment and treatment efficacy. The ORR also demonstrated a significant improvement in both populations treated with ICI plus CT (OR 1.35, 95% CI). Subgroup analysis revealed that PD-L1 positive patients experienced a notable increase in PFS after treatment with ICI plus CT (age 18–64 years: HR 0.71, 95% CI). Thus, the combination of ICI and CT significantly improved PFS and OS in both ITT and PD-L1-positive patients compared to CT. Additionally, patients without prior chemotherapy exposure exhibited a longer PFS, highlighting the efficacy of ICI plus CT, particularly as an early line treatment strategy for metastatic TNBC. Random-effects models were used to estimate the pooled HRs and odds ratios (ORs) with 95% confidence intervals (CIs).

Recent randomized controlled trials (RCTs) have yielded variable outcomes in the field of oncology. To further investigate, Villacampa et al. [35] conducted a systematic review and meta-analysis [38] to assess the efficacy of ICI combined with CT compared with CT alone in untreated TNBC patients. Three RCTs with 2400 TNBC patients were included in the study, showing that in patients with PD-L1-positive tumors, the addition of ICI significantly improved PFS (HR 0.67; 95% CI) and demonstrated an increased OS (HR 0.79; 95% CI). Additionally, CT-naïve patients showed greater PFS from ICI therapy (HR 0.53; 95% CI) than those who had received prior chemotherapy in the neoadjuvant/adjuvant setting (HR 0.81; 95% CI), proving that ICI plus CT significantly increases PFS in PD-L1-positive TNBC patients, with greater benefits seen in those without prior chemotherapy exposure.

Another meta-analysis included randomized clinical trials (RCTs) conducted by Zhang et al. [39], comparing immunotherapy with chemotherapy in patients with advanced TNBC, with a total of 3183 patients across six studies. The combination of PD-1/PD-L1 inhibitors with chemotherapy significantly prolonged PFS with a hazard ratio (HR 0.82; 95% CI) in patients with unresectable, locally advanced, or metastatic TNBC. However, no significant improvement in OS was observed compared with chemotherapy alone. Patients with locally advanced or metastatic TNBC effectively enhances PFS in PD-1/PD-L1 inhibitor-based immunotherapy. Following the Preferred Reporting Items for Systematic Reviews and Meta-Analyses (PRISMA) guidelines, pooled HR and OR were estimated using a random-effects Bayesian network meta-analysis conducted by Liang et al. [40]. A total of 4589 patients with TNBC in eight RCTs with ICI therapy showed a trend toward improved PFS and OS compared to chemotherapy alone. According to the Bayesian ranking profiles, pembrolizumab plus chemotherapy demonstrated the greatest survival benefit among TNBC treatment regimens. Atezolizumab plus chemotherapy also provided superior survival outcomes in a subgroup analysis of patients with PD-L1-positive status, despite the limited OS benefit observed in some studies.

ICI therapies in TNBC when combine with chemotherapy demonstrated improved PFS particularly in PD-L1-positive patients. Meta-analyses confirm that ICI plus chemotherapy enhances OS and ORR especially in administered at in early treatment and in chemotherapy-naïve patients.

## 3. Key Strategies to Enhance ICI Efficacy

ICIs have revolutionized cancer treatment across multiple malignancies; however, their efficacy remains limited in a significant proportion of patients. Researchers are exploring combination strategies to address these challenges and enhance the efficacy of ICIs. These cancer treatments include the combination of ICI with other immunotherapies, targeted therapies, chemotherapies, and radiation therapy. For instance, dual checkpoint blockade, such as the combination of anti-PD-1 and anti-CTLA-4 antibodies, has shown improved outcomes in several cancer types [61]. Adoptive T-cell therapy (ACT) involves the expansion and reinfusion of a patient’s own T-cells to activate cancer cells [62]. Another is CAR-T cell therapy, which genetically modifies a patient’s T-cells to express receptors that are specific to cancer antigens. mRNA vaccines encode tumor-specific antigens and immune-stimulating molecules to activate the immune system to target and eliminate cancer cells [63].

Oncolytic viruses selectively infect and lyse tumor cells, and when combined with radiotherapy or chemotherapy, they enhance treatment of metastatic tumors due to their tumor-specific infectivity [64,65]. Additionally, chemotherapy and radiotherapy can synergize with ICIs by promoting immunogenic cell death and enhancing tumor antigen presentation.

Table 3 summarizes the latest research on combination strategies aimed at improving ICI effectiveness in cancer therapy. Although ICIs have transformed the cancer landscape, their efficacy remains limited in some patients. Numerous studies have explored various synergistic approaches to enhance response rates, overcome resistance, and expand the therapeutic benefits of ICIs in multiple cancer types.

In a study by Kim et al. [66], ICI combined with platinum-based chemotherapy was effective in treating advanced or recurrent endometrial cancer (EC). A randomized controlled trial with 2335 patients found that combined therapy significantly prolonged PFS of 0.70 and OS 0.75 with 95% CI compared to chemotherapy alone. Patients with deficient mismatch repair (dMMR) experienced the most pronounced benefits in PFS 0.33 and OS 0.37 with 95% CI, highlighting the predictive role of MMR status. This study underscores the effectiveness of ICIs in enhancing chemotherapy efficacy by integrating personalized EC treatment strategies into the treatment regimen.

### 3.1. Anti-Angiogenesis

Clinical trials have explored several promising combination therapies to enhance the efficacy of ICIs in various cancer types. IMpower150 trial evaluated the combination of an anti-PD-L1 agent (atezolizumab) and anti-VEGFA therapy (bevacizumab) in patients with advanced NSCLC. This study demonstrated improved OS compared with chemotherapy alone [67]. Similarly, the JVDF trial—a non-randomized, multicohort, open-label, phase 1a/b study—evaluated the combination of the anti-VEGFR2 agent ramucirumab with the anti-PD-1 therapy pembrolizumab in NSCLC and gastric cancer (GC), revealing a significant improvement in OS compared to chemotherapy. Furthermore, the combination of ramucirumab and pembrolizumab enhances OS and response rates in patients with NSCLC [68,69].

In another study, the administration of anti-CTLA4 in combination with interferon-alpha (IFN-α) during the early treatment phases was associated with a reduction in T-cell clonality in the bloodstream, correlating with improved OS and PFS [90,91]. Silk et al. [91] revealed that high-dose anti-CTLA4 and IL-2 therapy demonstrated that granzyme B levels and the HLA-DM/HLA-DO ratio are reliable blood biomarkers for predicting treatment response, which is associated with decreased tumor size.

Combining human telomerase reverse transcriptase (hTERT) vaccination with anti-CTLA4 therapy led to an elevated tumor IFN-γ signature, while blood biopsies showed T-cell receptor (TCR) clones enriched by vaccination—indicating a robust immune response. Furthermore, extended PFS in patients who received a combination of anti-PD1 and a tailored neoantigen vaccine (NEO-PV-01) showed an increased TCR clonal repertoire and consistent TCR profiles. A strong correlation has been observed between the existence of antigen-experienced T and B cells and TCR repertoire features [92,93].

Yap et al. [84] evaluated anti-ICOS monotherapy or its combination with anti-PD-1 therapy and found that elevated ICOS-high CD4+ T-cell counts were associated with better clinical outcomes, suggesting ICOS as a potential biomarker for treatment response [94,95]. To improve the effectiveness of immunotherapy, enhancing the cytotoxicity of immune effector cells through the use of anti-angiogenic drugs will facilitate targeted delivery to tumor sites [96].

However, the efficacy of antibodies that specifically target CTLA4 and PD-1/PD-L1 has not been consistently beneficial in all patient groups. Therefore, researchers must explore other immune checkpoints as prospective ICI targets. Notably, LAG3, TIM3, and TIGIT are among the most promising immunotherapies being evaluated in clinical trials for HCC. Preliminary findings from clinical studies investigating inhibitors of these new targets have shown the potential for better outcomes in patients who do not respond to conventional ICI treatments [97].

### 3.2. Biomarker-Guided Therapy

Another emerging strategy to enhance ICI efficacy is biomarker-guided therapy, which customizes treatment based on the specific molecular characteristics of a patient’s tumor. Serum lactate dehydrogenase (LDH) was among the first clinically approved biomarkers used to predict the prognosis of metastatic melanoma based on baseline clinical characteristics. Studies show that elevated LDH levels serves as a negative prognostic factor for melanoma, regardless of the treatment administered. Patients with high LDH levels generally exhibit poorer overall survival (OS) than those with normal LDH levels, making LDH a valuable biomarker for disease staging and progression [98]. Currently, biomarkers such as PD-L1 expression, TMB, and microsatellite instability (MSI) are widely recognized as potential indicators of ICI response. Increased PD-L1 expression on tumor cells is frequently linked to an enhanced response to PD-1/PD-L1 inhibitors, making it a key factor in selecting patients for therapy. Other factors, including immunosuppressive components in the TME, such as regulatory T cells (Tregs) and myeloid-derived suppressor cells (MDSCs), contribute to ICI efficacy by weakening the immune response in cases of high PD-L1 expression. Additionally, genetic alterations influence tumor sensitivity to ICI, such as mutations in interferon-gamma pathway genes, which potentially limit its effectiveness despite elevated PD-L1 levels [99].

In a study by Ma et al. [70] combined therapy in HER2-negative advanced gastric cancer (GC) exploring predictive biomarkers for treatment. The study analyzed 190 patients with histologically confirmed adenocarcinoma (stage IV, HER2-negative, pMMR) and revealed an ORR of 61.9%, DCR of 96.8%, median PFS of 9 months, and OS of 27 months. Biomarker analysis shows that HER2 expression levels were significantly associated with treatment efficacy. Patients with HER2 ‘2+’ expression had longer PFS than those with HER2 ‘1+’ expression, suggesting that HER2 expression in HER2-negative GC may be a relevant predictive factor for trastuzumab efficacy. Furthermore, circulating tumor DNA (ctDNA) testing has demonstrated specific genetic alterations related to survival. Additionally, patients with SNVs in SMARCA4, RASA1, BRCA2, TP53, and STK11 and CNVs in MYC and EXT1 exhibited significantly shorter PFS and OS, indicating their potential as prognostic biomarkers for PFS and OS. These findings emphasize the critical role of biomarker-guided therapy in optimizing treatment efficacy for gastric cancer [98].

Moreover, while ICIs have transformed cancer treatment of HCC and the therapeutic effect of ICI (programmed cell death (PD)-1/programmed death-ligand1 (PD-L)1 antibody), combined ICI therapies, tyrosine kinase inhibitor (TKI) and ICI + locoregional treatment, enhancing antitumor immune responses, only few patients benefit from these therapies [100]. To optimize treatment outcomes in patients with cancer, studies and experiments should delve into the importance of biomarkers, such as PD-L1 expression, TMB, MSI, and immune cell infiltration within the tumor microenvironment. These can potentially predict cancer patients who are more likely to respond to ICI therapy, thereby guiding more personalized treatment. Biomarker-guided therapy enables personalized treatment strategies to optimize ICI efficacy. Key biomarkers, including LDH, PD-L1, TMB, and MSI, help predict therapeutic response, disease progression, and overall survival, thereby enhancing treatment precision across various cancers.

### 3.3. Oncolytic Viruses

Oncolytic viruses represent a promising strategy for cancer treatment, as they selectively target and destroy tumor cells while sparing normal tissue. The FDA has approved several oncolytic viruses for clinical use, and interest in this approach has grows rapidly [101]. Oncolytic viruses can be used alone or in combination with other treatments, including radiotherapy, chemotherapy, immunotherapy, and cell-based therapies. Currently, adenovirus (Ad), herpes simplex virus (HSV), vaccinia virus (VV), reovirus, poliovirus, coxsackie virus (CV), Newcastle disease virus (NDV), vesicular stomatitis virus (VSV), myxoma virus, and certain senteroviruses are in use [65].

In a study conducted by Guan et al. [71] through a phase 2 clinical trial, STOMP evaluated the combined use of stereotactic body radiation therapy (SBRT) and in situ oncolytic virus therapy in treating metastatic non-small cell lung cancer (mNSCLC) from 28 patients who received intratumoral injections of ADV/HSV-tk (5 × 10^11^ vp) along with SBRT (30 Gy across 5 fractions), followed by pembrolizumab (200 mg IV). The study reported an objective response rate (ORR) of 33.3% and a clinical benefit rate (CBR) of 70.4%. Notably, 75% of immune checkpoint inhibitor (ICI)-refractory patients experienced clinical benefit. Patients demonstrated a prolonged response, with a median PFS of 7.4 months and median OS of 18.1 months. These results suggest that combining ADV/HSV-tk gene therapy with valacyclovir and SBRT may enhance the antitumor efficacy of pembrolizumab in a chemotherapy-free regimen for mNSCLC.

Yi et al. [72] evaluated the safety and effectiveness of H101(a novel oncolytic adenovirus) combined with nivolumab in patients with HCC who had not responded to previous systemic treatments. Eighteen of the 21 screened patients received oncolytic virus (OV) pretreatment with intratumoral H101 injections (5.0 × 10^11^ vp/0.5 mL/vial, two vials per lesion), demonstrating an ORR of 11.1% with two cases of partial response (PR) and five cases of stable disease (SD). Extrahepatic administration was associated with favorable responses and a DCR of 38.9%. six-month survival rate was 88.9%, with a median PFS of 2.69 months and an OS of 15.04 months. Notably, local H101 injections appeared to reverse resistance to immune checkpoint inhibitors, resulting in prolonged OS exceeding 2.5 years in PR cases with low α-fetoprotein levels. Yi et al. [72] suggests that combining H101 with nivolumab could be a promising therapeutic approach for refractory advanced HCC.

Previous studies have demonstrated that vesicular stomatitis virus (VSV-S) exhibits superior anticancer activity compared to wild-type VSV in a subcutaneous TNBC mouse model. Tang et al. [102] evaluated the therapeutic effects on TNBC using an orthotopic mouse model with an anti-PD-1 antibody for treating metastasis. Neoadjuvant VSV-S treatment significantly reduced lung metastases after primary tumor resection and improved overall survival. The efficacy of immunotherapy is heavily dependent on the TME of the TNBC. However, a study treating a mouse model with Smac-armed VSV-S changed the TME. In a TNBC lung metastasis model, pulmonary administration of VSV-S significantly enhanced the efficacy of ICI therapy, suggesting that combining oncolytic virus therapy with ICI treatment holds great potential for improving therapeutic outcomes in TNBC. Recent clinical and preclinical studies have shown that combining oncolytic viruses with ICI can improve treatment responses across various cancer types, including mNSCLC, HCC, and TNBC. This highlights the potential of oncolytic viruses to overcome resistance to ICIs and improve patient outcomes.

### 3.4. Cancer Vaccines

mRNA-4157 (V940) is a personalized neoantigen therapy targeting up to 34 tumor-specific neoantigens in each patient, which aims to stimulate T-cell responses and enhance antitumor activity. Neoantigens can trigger antitumor T-cell responses, making them promising targets for cancer therapies (Figure 2). However, tumor mutations and their associated antigen-presenting molecules vary among patients [103].

In study conducted by Gainor et al. [73] provided mechanistic insights into the immunogenicity of mRNA-4157 by analyzing T-cell responses to neoantigens in the first-in-human, phase 1 KEYNOTE-603 trial. Patients with resected NSCLC (1 mg mRNA-4157) and resected cutaneous melanoma (1 mg mRNA-4157 + 200 mg pembrolizumab) were evaluated for safety, tolerability, and immune response. mRNA-4157 monotherapy consistently induced de novo T-cell responses and enhanced pre-existing responses to targeted neoantigens, whereas mRNA-4157 + pembrolizumab sustained neoantigen-specific T-cell responses and promoted the expansion of cytotoxic CD8 and CD4 T cells. However, patients reported experiencing at least one treatment-emergent adverse event, with no grade 4/5 toxicities or dose-limiting side effects observed. Gainor et al. [73] demonstrated the safety and immunogenicity of mRNA-4157 as an adjuvant monotherapy or in combination with pembrolizumab.

In an open-label randomized phase IIb adjuvant study, the addition of mRNA-based individualized neoantigen therapy to PD-1 blockade enhanced clinical outcomes in the adjuvant treatment of high-risk resected melanoma compared with PD-1 blockade alone. Patients with IIIB–IV melanoma were randomly assigned to receive either combination therapy or pembrolizumab monotherapy. A total of 157 patients were assigned to the mRNA-4157 plus pembrolizumab combination therapy group, with 107 receiving combination therapy and 50 receiving pembrolizumab monotherapy. The findings showed a longer recurrence-free survival in the combination group than in the monotherapy group (HR 0.561), with lower rates of recurrence or death (22% vs. 40%). At 18 months follow-up, recurrence-free survival was 79% compared to 62% with monotherapy, suggesting that adjuvant treatment with mRNA-4157 plus pembrolizumab extends recurrence-free survival compared to pembrolizumab monotherapy in patients with resected high-risk melanoma [104].

In a cohort study conducted by Kyr et al. [105] in pediatric patients who received personalized DC vaccines, an overall survival of 7.03 years was demonstrated. Disease control following DC vaccination was observed in 53.8% of the patients. It also revealed improved outcomes in patients who received the vaccine more than two years post-diagnosis (HR = 0.53). Additionally, a strong synergistic effect was observed when DC vaccines were combined with ICIs (HR = 0.40), and a positive trend was also observed with metronomic cyclophosphamide and/or vinblastine (HR = 0.60). This study emphasizes the potential of personalized DC vaccines as safe and effective components of individualized pediatric cancer therapy.

Another vaccine under investigation for TNBC has demonstrated promising results, with over 70% of patients exhibiting immune responses. Findings from a phase 1 clinical trial evaluated the α-lactalbumin vaccine in three cohorts. Cohort 1a included 21 women previously treated for TNBC, cohort 1b comprised two women at high genetic risk of TNBC undergoing prophylactic mastectomy, and cohort 1c consisted of three TNBC patients receiving the vaccine alongside pembrolizumab (Keytruda). In cohort 1a, strong immune responses were observed, similar to those in cohort 1b. Meanwhile, cohort 1c showed no adverse events with the combination therapy. Confirmed immune response data were also recorded. These three cohorts demonstrated a favorable safety profile, with injection-site irritation being the most common adverse event reported. No major systemic adverse effects were observed, even when administered with pembrolizumab, supporting the potential use of the α-lactalbumin vaccine as immunotherapy for TNBC [106]. Cancer vaccines, including mRNA-4157 (V940), DC vaccines, and tumor-specific antigens such as α-lactalbumin, are emerging immunotherapeutic strategies that offer hope to cancer patients. These approaches demonstrate the ability to stimulate neoantigen-specific T-cell responses, improve recurrence-free survival, and synergize with ICIs.

### 3.5. Radiotherapy and ICI Synergy

An increase in PD-1^+^ CD8^+^ T cells and PD-L1^+^ CD8^+^ T cells has been identified as a hallmark in patients receiving combination therapy with anti-PD-1 and radiation therapy, as indicated by elevated Ki67 expression prior to treatment. This implies a rapid multiplication of immune cells that has a substantial impact on therapy responses. In contrast, the presence of PD-1/PDL-1 on non-proliferating (Ki67−) CD8+ and CD4+ T cells was linked to worse results [107]. Combination therapy with radiation and anti-CTLA-4 leads to increased levels of ICOS^+^ CD4^+^ and CD8^+^ T cells; however, only CD8^+^ T-cell elevation has been directly linked to improved PFS [95,108]. These findings emphasize the importance of certain immune cell markers, such as Ki67 and ICOS, in predicting the effectiveness of combination ICI and radiation therapy. This synergistic approach has been shown to promote systemic T-cell activation, including upregulation of ICOS and PD-1 on CD8^+^ T cells. Melanoma patients with brain metastases showed a positive response to the combination treatment, resulting in an increase in activated memory CD8+ T cells and ICOS+ CD8+ T cells. Furthermore, improved tumor-specific antigen reactivity was associated with higher levels of interferon-gamma (IFN-γ) produced by T cells [90].

The combination of radioembolization and ICI has emerged as a promising strategy for enhancing antitumor response in HCC. Recent advancements in liver-directed therapies, such as yttrium-90 (Y90) radioembolization, alongside ICI treatments, such as atezolizumab, bevacizumab, durvalumab, and sorafenib, have expanded the therapeutic options for patients with HCC [109]. Radioembolization delivers targeted radiation to liver tumors, resulting in tumor cell death and TME modulation. It enhances tumor antigen release and potentially increases the effectiveness of ICIs that block inhibitory pathways, such as PD-1/PD-L1 and CTLA-4, and restores T-cell activity against tumor cells. This combination integrates local tumor control with systemic immune activation, resulting in a synergistic antineoplastic effect [110].

A retrospective analysis was conducted by Garcia-Reyes et al. [74] in 44 patients with HCC who underwent yttrium-90 (90Y) radioembolization with initial ICI or TKI therapy using Kaplan–Meier analysis, which demonstrated a higher ORR in patients receiving the combination of 90Y and ICI therapy (89.5% vs. 36.8%) and DCR of 94.7% vs. 63.2% than those yielded by TKI therapy and 90Y for the treatment alone. Based on the mRECIST and iRECIST criteria, the ORR was 78.9% and 36.8%, and the DCR was 94.7% and 63.2%, respectively. However, there was no statistically significant difference in the median PFS (8.3 vs. 4.1 months) or OS (15.8 vs. 14.3 months) between groups.

Low-dose radiation therapy (LDRT) transiently induces inflammation within tumors, rendering them more receptive to immunotherapy. This is achieved by altering immunosuppressive factors, thereby enhancing the infiltration and activity of immune effector cells against cancer. In preclinical models, LDRT reduced tumor burden, improved survival rate, and resulted in a remarkable 40% increase in the complete response rate. In immunologically cold melanoma models, low-dose whole-brain radiation therapy (LD-WBRT) 4 Gy single fraction reduced intracranial tumors and improved survival rates, while a high-dose radiation-based in situ vaccination (ISV) in 12 Gy single fraction immune-cytokines and anti-CTLA-4 effectively eradicated primary flank tumors. Therefore, LDRT enhances the effectiveness of immunotherapy-based treatments when combined with ICI, CAR-T cell therapy, cytokine therapy, tumor-infiltrating lymphocyte (TIL) therapy, oncolytic virus therapy, and cancer vaccines, resulting in synergistic effects. Furthermore, these combination therapies not only enhance tumor control and improve OS but also overcome tumor recurrence and resistance to standard therapies [111]. Combination therapy with ICI offers a synergistic strategy to enhance antitumor immune responses by promoting T-cell activation, improving tumor antigen presentation, and altering the TME to favor immune infiltration. Clinical and preclinical studies have demonstrated that markers such as Ki67, ICOS, and IFN-γ can help predict treatment response.

### 3.6. ICIs with JAK Inhibitors

Recent clinical trials by Zak et al. [75] and Mathew et al. [76] investigated the efficacy of combining therapeutic strategies with ICI. Their studies demonstrated that combination therapies yielded superior clinical responses compared to ICI monotherapy in patients with relapsed or refractory Hodgkin lymphoma and metastatic NSCLC. ICIs enhance the immune system’s ability to target and destroy cancer cells, representing a significant breakthrough in cancer treatment. However, not all patients respond effectively to ICI, prompting researchers and clinicians to explore combination therapies to improve its effectiveness, one of which involves combining ICI with Janus kinase (JAK) inhibitors (JAKi) to enhance therapeutic outcomes [112].

JAK inhibitors are small molecules that target the JAK family of enzymes, which are key regulators of cytokine and growth factor signaling in hematopoiesis, immune regulation, inflammation, and neuropathology. Blocking JAK can modulate immune responses, making it effective in the treatment of autoimmune diseases. In most autoimmune conditions, JAKi reduce T-cell activity, in contrast to the TME, which is characterized by excessive T-cell activation; thus, JAKi are beneficial in combination with conventional therapies. ICI enhance the immune system’s ability to recognize and attack cancer cells, whereas JAKi modify the immune microenvironment by mitigating factors that suppress the response rather than directly enhancing cytotoxic immune cells, JAKi reduce immunosuppressive signaling and reset chronic inflammatory pathways within the TME, thus improving the efficacy of ICI therapies.

Currently, clinical research has examined the efficacy of combining JAK inhibitors with ICIs to enhance treatment outcomes, such as in malignancies that are resistant to traditional immunotherapy. Collaborative studies with researchers at the University of Minnesota to conduct a trial of the JAK inhibitor ruxolitinib (Jakafi) combined with nivolumab in patients with Hodgkin lymphoma who had previously shown no therapeutic response to ICI revealed a cancer shrinkage of 53% in patients who received the combination therapy, while 46% exhibited no indications of disease progression two years after initiating the therapy. Furthermore, this combination method enhanced the functionality of T cells and myeloid cells, which is important for the immunological response against cancer. This combination presents new innovations and opportunities for the treatment of immunotherapy-resistant malignancies by reducing immune evasion mechanisms and enhancing OS and PFS in patients with restricted therapy alternatives [113,114].

Chronic inflammation mediated by type-one interferon (IFN-I) can lead to immunosuppression. Matthew et al. [76] demonstrated that the administration of Janus kinase 1 (JAK1) inhibitor itacitinib after pembrolizumab (anti-PD-1) enhanced immune function and antitumor responses in mice, achieving a 67% response rate in a phase 2 clinical trial for metastatic NSCLC. A trial that included 21 patients with treatment-naïve NSCLC showed that the median PFS was nearly 2 years, versus 6.5 to 10.3 months in monotherapy trials. Importantly, patients who initially failed PD-1 blockade showed improved responses after the addition of itacitinib, suggesting that JAK inhibition can reverse poor immune function by modulating CD8^+^ T-cell differentiation. Patients with chronic inflammation unresponsive to itacitinib experienced continued CD8 T-cell terminal differentiation and disease progression, suggesting that JAK inhibition enhances the effectiveness of anti-PD-1 immunotherapy by modulating T-cell differentiation dynamics. JAK inhibitors modulate the TME by reducing chronic inflammation and T-cell exhaustion, thereby enhancing the efficacy of ICIs. Clinical trials have shown remarkable outcomes, including improved tumor response, PFS, and OS. These findings support the potential of ICI–JAK inhibitor combinations to overcome immune resistance and broaden the scope of effective immunotherapy for cancer.

### 3.7. Microbiome Modulation

Clinical experiments and reviews have demonstrated the influence of the gut microbiota on the efficacy and resistance to ICIs, and whether its modulation can enhance responses to immunotherapy [115].

The association between the gut microbiome and the effectiveness of ICI therapy has prompted investigations into the causal role of the gut microbiota in enhancing its responses. The growing evidence linking gut microbiota composition to ICI therapy outcomes has prompted investigations into its causal role in enhancing antitumor responses. Most findings indicate that the microbiome primarily boosts ICI response by modulating the body’s antitumor immune responses, while some studies suggest that microbial metabolites may directly kill tumor cells [116,117]. Short-chain fatty acids (SCFAs), bile acids, and hippurate are metabolites produced by the gut microbiota that significantly regulate antitumor immune responses. These metabolites can cross the intestinal barrier and influence both local and systemic immune responses against tumors, thereby reshaping the TME and boosting ICI effectiveness. Thus, modulating the antitumor immune response through the microbiota-metabolite-immune axis is a key mechanism by which the gut microbiota can promote improved outcomes in ICI therapy [115,118,119]. Table 4 summarizes the influence of gut microbiota on ICI efficacy. It highlights the mechanism of action, including immune cell modulation, metabolite production, enhancement of tumor antigen presentation, and gut–brain axis signaling.

The processes by which the gut microbiota impacts immunological homeostasis include the regulation of immune cell activity, generation of metabolites that impact inflammation, and augmentation of tumor antigen presentation [120]. Although the interaction between gut microbiota and ICI is complex and complicated. The gastrointestinal microbiota not only directly regulates the activity and function of immune cells to affect the effectiveness of immunological cytokines but also modulates the immune system via the gut–brain axis, thereby influencing the efficacy of ICIs against malignancies [121]. However, gut microbiota dysbiosis can lead to tumor resistance to ICI therapy [122].

**Table 4 cancers-17-01408-t004:** Gut Microbiota Influence on ICI Efficacy and Mechanistic Pathways.

Key Findings	Mechanism of Action	Effect on ICI Therapy	Reference
Gut microbiota modulates GM-CSF via gut–brain axis	↑ ROS in immature myeloid cells; ↑ MDSC suppression of T cells	Enhances ICI response by weakening immunosuppressive barriers	[77,123]
Antigenic similarity between *Enterococcus hirae* proteins and tumor antigens	Stimulates CD8^+^ T cells via β-type 4 antigens	Boosts PD-1 blockade efficacy through cross-reactivity	[124]
Gut microbiota synergizes with anti-CTLA-4 therapy in GBM	↑ IFN-γ production; ↑ microglial phagocytosis via CD4^+^ T cell modulation	Improves ICI response and tumor clearance in glioblastoma	[78]
Microbiota modulates FMT and PD-1 therapy response	↑ MAIT and CD56^+^CD8^+^ T cells; CD74 + GZMK → ↑ HLA-II expression	Enhances ICI through improved immune cell infiltration and activation	[79]
Gut-driven bile acid metabolism attracts CXCR6^+^ NKT cells in HCC	CXCL16 from liver sinusoidal endothelial cells → NKT recruitment	Increases ICI efficacy in HCC; probiotics aid liver recovery and reduce toxicity	[125]

Collective studies underscore the multifaceted role of the gut microbiota in modulating host immune responses and improving ICI therapy outcomes through both immune and metabolic pathways. GM-CSF—Granulocyte-Macrophage Colony-Stimulating Factor; ROS—Reactive Oxygen Species; MDSC—Myeloid-Derived Suppressor Cell; GBM—Glioblastoma Multiforme; FMT—Fecal Microbiota Transplantation; MAIT—Mucosal-Associated Invariant T Cells; GZMK—Granzyme K; HLA-II—Human Leukocyte Antigen Class II; CXCR6—C-X-C Motif Chemokine Receptor 6; CXCL16—C-X-C Motif Chemokine Ligand 16. ↓ indicating a reduction; ↑ indicating an increase.

Finally, oral intake of microbiome capsules, a new probiotic formulated from fecal bacteria from healthy donors or HCC patients who responded to ICI therapy, has the potential to improve immunotherapy effectiveness in HCC patients with inadequate immune responses by modulating gut microbiota. Moreover, probiotics aid in mitigating the side effects of anticancer treatment, thus supporting liver function recovery [125]. The microbiome influences both local and systemic antitumor responses through various mechanisms, including modulation of immune cells, production of immunoregulatory metabolites, enhancement of tumor antigen presentation, and gut–brain axis signaling, thereby enhancing the effectiveness of ICI.

### 3.8. Checkpoints Inhibitors: LAG3, TIM3, and TIGIT

Re-expression of alternative immune checkpoints, such as Lymphocyte Activation Gene-3 (LAG-3), T-cell immunoglobulin and mucin domain 3 (TIM-3), and T-cell immunoglobulin and (ITIM) domain (TIGIT), can lead to resistance, which acts as an escape mechanism for tumors after initial ICI treatment [126]. Emerging immune checkpoints, such as LAG3, play crucial roles in downregulating T-cell activation [127]. Renal clear cell carcinoma (KIRC), kidney renal papillary cell carcinoma (KIRP), and several other malignancies, as LAG3 expression has been associated with poor prognosis due to various cancers. Emerging checkpoints like LAG-3 play a pivotal role in downregulating T-cell activation, and their expression has been associated with poor prognosis in various malignancies, including renal clear cell carcinoma (KIRC) and kidney renal papillary cell carcinoma (KIRP) [128,129].

An ongoing clinical trial by Majim et al. [80] is investigating inhibitors that target elevated ICI in NSCLC, including LAG-3, TIGIT, TIM-3, and sialic acid-binding Ig-like lectin 15 (Siglec-15). Patients resistant to anti-PD-1 therapy showed an ORR of 8.3% and a DCR of 33% in the TACTI-002 study, which combined eftilagimod alpha (a soluble LAG-3 inhibitor) with pembrolizumab [130].

Owing to its role in immune regulation, TIM3 inhibition presents another promising therapeutic target for monotherapy and in combination with PD-1 blockade therapy. Combining TIM3 and PD-1 inhibitors in preclinical studies has demonstrated enhanced immune restoration, as exhausted T cells in the tumor microenvironment commonly express both TIM3 and PD-1. Targeting these pathways has a synergistic effect, helping to reverse T-cell exhaustion and restore antitumor immune responses. The combination of TIM3 and PD-1 has been supported by evidence showing improved immune activity and potential efficacy in overcoming resistance to single-agent PD-1 inhibitors [128,131].

Dual targeting of LAG-3 and TIGIT or triple inhibition of LAG-3, PD-1, and Tim-3 reduces tumor growth. In a study by Mimura et al. [81], blocking both LAG-3 and TIGIT resulted in complete tumor regression in a significant number of mice and improved survival rates. In addition, the combination of LAG-3, PD-1, and Tim-3 enhances B-granzyme production by CD8+ tumor-infiltrating lymphocytes (TILs) and increases the cytotoxic activity of T lymphocytes against cancer cells [132].

Similarly to other checkpoints, such as PD-1 and LAG3, TIGIT is expressed on exhausted T cells and natural killer (NK) cells, and its re-expression after initial therapy allows tumors to evade immune responses. Tumor cells can continue to suppress immune activation when TIGIT is inhibited, thereby limiting the efficacy of existing ICI therapy. Ongoing research is determining how TIGIT inhibitors can be used in combination with PD-1/PD-L1 therapies to overcome this resistance and restore antitumor immune responses [128]. It is abundantly expressed on activated NK cells and is initially associated with enhanced antitumor activity. However, TIGIT environments can reduce NK cell function. The use of anti-TIGIT antibodies potentially boosts the antitumor efficacy of PM21-NK cells. The combination of adoptive PM21-NK cells with anti-TIGIT antibodies revealed higher efficacy in ICI treatment; however, it should be further investigated in preclinical studies and clinical trials for the treatment of lung tumors [133].

Administering anti-LAG-3 antibodies alongside PD-1 blockade has demonstrated improved ORR, prolonged PFS, better treatment tolerability, and reduced mortality risk. However, further research is required to elucidate the specific therapeutic benefits of this approach. Re-expression of alternative immune checkpoints, such as LAG-3, TIM-3, and TIGIT, contributes to tumor immune escape and resistance to PD-1/PD-L1 blockade. Targeting these inhibitory pathways enhances immune activation, restored T-cell function, and improved tumor control.

### 3.9. Metabolic Reprogramming in Tumors

Tumors undergo metabolic reprogramming, including increased glycolysis and hypoxia, which create an immunosuppressive environment for tumor growth and promote tumor progression. It modifies metabolic processes to support the growth, survival, and continuous proliferation of cancer cells [82]. Such metabolic reprogramming increases the uptake of glucose and its conversion into lactate through fermentation, regardless of the full function of mitochondria, a commonly metabolic shift called the Warburg Effect, which allows cancer cells to thrive and maintain their aggressive growth [83] and the accumulation of lactic acid, resulting in the dysfunction of immune cells, such as T cells, within the TME [82].

Metabolic reprogramming contributes to the formation of an acidic, hypoxic, and nutrient-depleted TME, favoring the rapid production of ATP through aerobic glycolysis. It affects T-cell function, impairing their ability to mount effective antitumor immune responses. Thus, they not only support tumor growth but also contribute to immune evasion by weakening the activity of immune cells, such as T-cells, in the TME [84,85].

The metabolic diversity of tumor cells is evident in lung cancer. In NSCLC, Transforming Growth Factor Beta (TGF-β) regulates glycolysis. Under normal oxygen conditions, TGF-β suppresses glycolysis, whereas hypoxic environments (in vitro and in vivo) enhance glycolytic activity in tumor cells. This occurs due to the interaction of Hypoxia-Inducible Factor 1-alpha (HIF-1α) and the MH2 domain of phosphorylated Smad3, which results in alterations in Smad protein interactions at low oxygen levels [86].

Metformin, a metabolic modulator of lipid metabolism, suppresses oxygen consumption in tumor cells and alleviates intratumoral hypoxia. When combined with a PD-1 antibody, metformin enhances T-cell function and promotes tumor clearance in melanoma mouse models. Treatment outcomes were demonstrated in melanoma patients treated with a combination of metformin and PD-1 checkpoint inhibitors, such as nivolumab or pembrolizumab. A higher response rate and OS were observed in patients with NSCLC who received metformin in combination with ICI therapy [87,88].

ICI are ligand-receptor systems that either stimulate or inhibit immune responses, and their function is influenced by hypoxic conditions [134]. It alters the expression of several immune-related molecules, such as PD-L1, human leukocyte antigen G (HLA-G), CD47, and the immune checkpoint V-domain Ig suppressor of T-cell activation (VISTA), thereby creating an immunosuppressive TME. This facilitates tumor immune evasion, allowing cancer cells to escape immune surveillance and promoting tumor progression [135]. Cancer metabolic reprogramming is a novel characteristic of carcinogenesis and antitumor immune response. Similarly to cancer cells, immune cells in the tumor microenvironment experience significant metabolic reprogramming, which greatly influences antitumor immune response. These metabolic shifts are key contributors to both primary and acquired resistance to ICIs, such as glycolysis and hypoxia in the TME, which prevent proper T-cell activation and effector function, and through ICIs, resulting in effective tumor eradication [136].

Another study suggested that enhancing the effectiveness of ICI therapy targets glucose metabolism. Single-cell RNA sequencing (scRNAseq) analysis revealed that the glucose transporter Glut14 (SLC2A14) was highly expressed in T cells from patients who responded to PD-1 blockade. Increased Glut14 expression from flow cytometry analysis of circulating T cells in responders indicated that the modulation of glucose uptake improved ICI efficacy. The FDA-approved glycolysis inhibitor dimethyl fumarate (DMF) demonstrated enhanced antitumor responses against 4T1 tumors and exhibited a synergistic effect when combined with anti-PD-1 therapy. Monocarboxylate transporter 1 (MCT1), a lactate metabolism enzyme expressed in regulatory T cells (Tregs) within the TME, facilitates lactate uptake and induces PD-1 expression in Tregs. Inhibition of MCT1 in Tregs improved the antitumor effect of anti-PD-1 therapy. Deletion of the transporter SLC16A1 in Tregs reduces tumor growth and enhances synergy with ICI therapy. Additionally, targeting lactate metabolism, lactate dehydrogenase A (LDHA), which disrupts the lactate pathway, improves immune responses to ICI therapy [87,137]. Tumor metabolic reprogramming contributes significantly to immune evasion by creating a suppressive TME that impairs T-cell function and reduces ICI efficacy. Targeting glucose and lactate metabolism, modulating hypoxia, and using metabolic drugs such as metformin and dimethyl fumarate restore immune activity and enhance ICI responses.

### 3.10. Glutamine Metabolism (GM)

Several inhibitors targeting glutamine metabolism have been developed, focusing on blocking glutamine utilization, inhibiting key enzymes involved in its metabolic pathway, or disrupting essential glutamine transport proteins. The combination of glutamine metabolism inhibitors and ICI has been shown to be effective in suppressing cancer progression. Clinical trials have revealed that oral glutamine supplementation in patients with cancer strengthens immune function, mitigates the side effects of chemotherapy and radiotherapy, and enhances treatment outcomes [138].

Under hypoxic conditions, cancer cells utilize alternative substrates for energy metabolism, such as L-glutamine. Chen et al. [135] reveal the L-glutamine plays in tumor cell proliferation as an alternative energy source for tumor metabolism. Furthermore, melanoma cells release significant quantities of PD-L1 in the form of exosomes to stimulate M2 polarization of tumor-associated macrophages (TAM) when the glutamate/cystine reverse transporter is blocked, resulting in resistance to ICI [139]. Best et al. [140] revealed that elevated glutamate levels in cancer cells hampered the response to ICI.

Moreover, oncogenic cells exhibiting aberrant glutamic acid decarboxylase (GAD) expression can produce γ-aminobutyric acid. To promote the growth of tumor cells and prevent the infiltration of CD8+ T cells, gamma aminobutyric acid (GABA) attaches to and triggers GABAB receptors to suppress GSK-3β and facilitate β-catenin signaling, resulting in resistance to anti-PD-1 [141,142]. Lactic acid reduced the production of IFN-γ by T cells in a mouse melanoma model. Elevated LDH expression is associated with decreased T-cell activation, suggesting that lactate accumulation within the TME contributes to immune response suppression and impairs key immune functions such as cytokine production and T-cell activation [143].

In a separate study, Morozumi et al. [144] investigated strategies to overcome TKI resistance in renal cell carcinoma (RCC) and the role of GM in restoring TKI sensitivity, demonstrating improved prognosis in RCC patients through combination therapies with TKIs and ICI. In this study, we established sunitinib (Su)- and cabozantinib (Cabo)-resistant RCC cell lines (786-O, Caki-1, and ACHN) to analyze the correlation between GM and VEGF signaling before and after the development of drug resistance. Our findings showed that GM and VEGF signaling were upregulated in all TKI-resistant cells. Glutaminolysis resulted in a 40–47% cell-killing effect in Su-resistant cells and 35–55% in Cabo-resistant cells (in vivo). Furthermore, GM inhibition significantly reduced angiogenesis, as demonstrated by CD31 immunostaining. VEGF signaling and VEGFR2 expression were downregulated, and PTEN was upregulated, suggesting that GM treatment disrupts the TME and restores TKI sensitivity. The potential of integrating GM inhibitors with ICIs enhances the efficacy of immunotherapy in various cancer types.

Another study by Ying et al. [89] explored the role of GM in HCC and developed a transcriptome-based GMScore to quantify GM activity. Despite the critical role of GM in HCC, a standardized method for assessing GM activity is lacking. Using transcriptome data from two independent HCC cohorts, The Cancer Genome Atlas (TCGA) and the International Cancer Genome Consortium (ICGC), they analyzed the expression of 41 GM-associated genes to construct and validate the GMscore. Moreover, several genomic and transcriptomic biomarkers were examined and predicted using the tumor immune dysfunction and exclusion (TIDE) algorithm for tumor response to ICI. The GMScore was significantly correlated with tumor stage, histological grade, alpha-fetoprotein (AFP) levels, and vascular invasion, and a higher GMScore was identified as an independent risk factor for OS in both cohorts. Furthermore, high GMScore tumors displayed enhanced cell growth and genetic stability, which were linked to poor OS in patients undergoing transcatheter arterial chemoembolization (TACE). High-GMScore tumors exhibited elevated immune checkpoint gene expression, increased Treg infiltration, and reduced M1 macrophage presence, resulting in an immunosuppressive TME. A high GMScore was associated with poor predicted responses to ICI, as validated in an ICI-treated melanoma cohort [145]. Therefore, GMScore serves as a robust prognostic indicator that can be used in a clinical framework.

The Glutamine Metabolism-Related Immune Index (GMII), with eight independent prognostic genes was developed to highly predict survival outcomes in bladder cancer. Patients were classified into low- and high-risk groups based on clinical features, somatic mutations, immune cell infiltration, chemotherapeutic response, and immunotherapeutic efficacy. Patients in the low-risk group demonstrated better responses to gemcitabine chemotherapy and ICI than those in the high-risk group. GMII offers potential therapeutic strategies not only for bladder but for all types of cancer [146]. Glutamine metabolism plays a pivotal role in tumor growth, immune suppression, and resistance to immune checkpoint inhibitors (ICIs). Targeting GM through inhibitors or clinical indices, such as GMScore, restores T-cell activity, overcomes ICI resistance, and improves treatment outcomes.

## 4. Immune-Related Adverse Events (irAEs)

While ICIs have revolutionized cancer immunotherapy, treatment resistance and immune-related adverse events (irAEs) remain a challenge, affecting multiple organs system [147]. Among the less commonly reported irAEs is the noninfectious uveitis that has been increasingly recognized among patients receiving ICIs. Retrospective cohort study in South Korea revealed cumulative uveitis incidence of 0.35% within one year of initiating treatment in patients with skin melanoma or lung cancer slightly higher than those receiving cytotoxic chemotherapy (0.33%) [148]. In addition, recent evidence shows ocular toxicities like uveitis, with skin melanoma patients showing a 0.7% cumulative incidence following ICI treatment [149]. Given the complexity of irAEs that highlight the unpredictability of immune-related responses, there is a growing need to optimize treatment selection and minimize adverse events, gearing toward a more personalized immunotherapy treatment. Rare cases of irAEs can occur, including ICI-induced type 1 diabetes mellitus (ICI-T1DM), as Guillain-Barré syndrome, myopathy, and myasthenia gravis [150]. Severe dermatologic irAEs range from rashes to Stevens-Johnson syndrome (SJS) and toxic epidermal necrolysis (TEN) [147,151]. Early detection and appropriate management of these irAEs, typically through immunosuppressive therapies, are critical to ensuring patient safety and treatment success.

## 5. Conclusions

This review underscores the transformative role of immune checkpoint inhibitors (ICIs) in cancer therapy, highlighting their ability to modulate the immune system to effectively target and eliminate tumor cells. PD-1, PD-L1, and CTLA-4 inhibitors have significantly improved OS and prolonged PFS in various malignancies, such as NSCLC, HCC, melanoma, and TNBC. Despite these successes, primary and acquired resistance continue to pose a major clinical challenge, leading researchers and clinicians to explore more combination strategies to enhance their effectiveness.

Several approaches are available for cancer treatment, as well as emerging therapeutic approaches that synergize ICIs with standard treatments such as radiotherapy, targeted therapy, microbiome modulation, metabolic reprogramming, oncolytic virus therapy, and cancer vaccines that are showing promising outcomes in overcoming resistance and improving patient prognosis. Radiotherapy, for instance enhances tumor antigen presentation and immune activation. Targeted therapies, such as BRAF/MEK inhibitors in melanoma and TKIs, optimize ICI responses by modulating the TME and gut microbiome, which are key factors that improve ICI responses with certain bacterial strains that enhance antitumor immune activity.

Metabolic reprogramming is a critical component of tumor immune evasion. Strategies targeting glucose and glutamine metabolism have shown potential for restoring ICI sensitivity and enhancing treatment response in resistant cancers. Another strategy for improving immune function and prolonging survival in relapsed or refractory cancers is the use of JAKi in combination with ICIs. Oncolytic virus therapy has also demonstrated potential in reshaping the immune landscape, particularly in melanoma, NSCLC, and HCC, by stimulating CD8+ T-cell activity and enhancing the ICI response. Cancer vaccines have emerged as a novel approach to boost tumor-specific immune response.

In addition, researchers are exploring biomarker-driven strategies to personalize ICI therapy and improve patient selection and treatment. Biomarkers such as PD-L1 expression, TMB, and MSI are increasingly used to predict ICI efficacy. Moreover, novel immune checkpoints, such as LAG-3, TIM-3, and TIGIT, are being investigated to overcome ICI resistance and further expand immunotherapy applications. Furthermore, ICI remains a cornerstone in modern and advanced oncology, but overcoming resistance and limitations through innovative combination therapies, such as precision immunotherapy and emerging immune-modulating strategies, will be essential for improving patient survival and advancing the next generation of cancer immunotherapy.

## Figures and Tables

**Figure 1 cancers-17-01408-f001:**
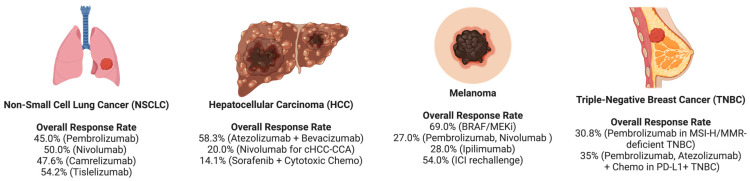
Overall response rates of major cancer types, such as NSCLC, HCC, melanoma, and TNBC, and ICI administration in the studies. ICI include pembrolizumab, nivolumab, atezolizumab, and bevacizumab, which emphasize the variability in response rates across different cancer types treated with ICIs (Created in BioRender. Valencia, M. (2025). https://BioRender.com/zw1wmv9, accessed on 16 April 2025).

**Figure 2 cancers-17-01408-f002:**
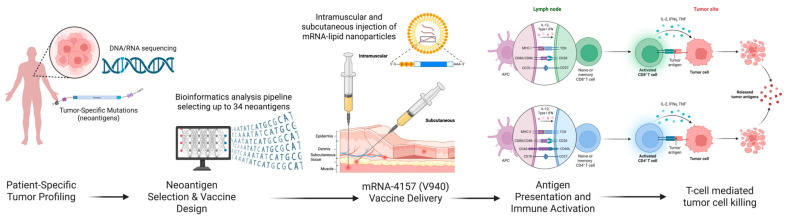
Schematic overview of the personalized mRNA-4157 (V940) vaccine strategy for neoantigen-driven cancer immunotherapy (Created in BioRender. Valencia, M. (2025). https://BioRender.com/jrtkrgt, accessed on 16 April 2025).

**Table 1 cancers-17-01408-t001:** Summary of ICI Clinical Outcomes.

Population and Design	Strategies Utilized	ORR	DCR	PFS (Months)	OS (Months)	Reference
Retrospective; n = 134	Long-term ICI monotherapy (≥18 months)	58.2% (overall); 33.3% (≥18 months)	83.3% (≥18 months)	10.6	Not reached (↑ if ≥18 months ICI)	[15]
Retrospective; n = 145	ICI + Chemotherapy or Anti-angiogenic	29.3%	85.4%	6.77	18.60	[16]
Meta-analysis; n = 2410	ICI + ICI, ICI + chemo, various combos	RR 1.82 (↑)	RR 1.41 (↑)	HR 0.83 (↑)	HR 0.90 (↑)	[17]
Retrospective; n = 143	ICI + TKIs, cross-line ICI	11.2%	72.7%	4.6	11.8	[18]
Real-world; n = 110	ICIs (monotherapy or with novel/anti-angiogenic agents)	-	75%	5.5	20.3	[19]
Real-world; n = 90	ICI-based therapy	36.7%	78.9%	4.9	13.9	[20]
Real-world; n = 244	First-line ICI monotherapy	-	-	7.0; 11.3 (if >3 months immunotherapy)	11.8; 15.4 (if >3 months immunotherapy)	[21]

n overview of clinical studies investigating the efficacy of ICI therapies across various cancer types and the strategies utilized. Studies report clinical outcomes and real-world data, including risk ratio (RR) for objective response rate (ORR), disease control rate (DCR), hazard ratio (HR), progression-free survival (PFS), and overall survival (OS). The arrows help highlight differences and trends in data between group such as (arrow down) indicate reduction or while (arrow up) indicate increase or improvement (e.g., OS, PFS, ORR).

**Table 2 cancers-17-01408-t002:** ICI Response Rate and Efficacy Metrics According to Cancer Type.

Cancer Type	ORR	PFS	OS	DCR	References
NSCLC	Pembrolizumab: 45.0%, Nivolumab: 50% Camrelizumab: 47.6%, Tislelizumab: 54.2%	Median PFS for ICI-based therapies: 9.5 months. Pembrolizumab: 9.6 months, Nivolumab: 9.2 months, Camrelizumab: 10.4 months, Tislelizumab: 10.3 months	Median OS for ICI monotherapy: 10.9 months, chemotherapy: 10.7 months, ICI combination therapy: 20.3 months. In PD-L1 ≥ 1%, OS improved in ICI (22.4 months) vs. ICI + chemo (10.7 months)	ICI-treated elderly patients achieved 75% DCR	[19,27,28],
HCC	Atezo + Bev: 58.3%, Dur + Tre: 0%, Sorafenib and Cytotoxic Chemo: 14.1%, ICIs for cHCC-CCA: 20%	Atezo + Bev: 2.9 months, Lenvatinib (2L): 4.0 months, Sorafenib (2L): 2.3 months, TKI + ICI: 5.4 months, cHCC-CCA (ICIs): 3.5 months, Sorafenib and Cytotoxic Chemo: 3.8 months	Atezo + Bev: 8.0 months, Lenvatinib (2L): 8.0 months, Sorafenib (2L): 6.3 months, TKI + ICI: 12.6 months, cHCC-CCA (ICIs): 8.3 months, Sorafenib and Cytotoxic Chemo: 10.6 months	Atezo + Bev: 87.5%, Dur + Tre: 62.5%	[29,30,31,32]
Melanoma	BRAF/MEKi: 69%, Nivolumab: 27%, Nivolumab + Ipilimumab: 28% ICI rechallenge: 54%	BRAF/MEKi: 14.7 months, Anti-PD-1: 5.4 months, PD-1/CTLA-4: 5.8 months, ICI rechallenge: 21 months, Pembrolizumab (previously treated): 3.9 months, Pembrolizumab (naïve): 2.3 months	BRAF/MEKi: 34.6 months, Anti-PD-1: 37.0 months, Pembrolizumab (previously treated): 19.0 months, Pembrolizumab (naïve): 6.8 months, ICI rechallenge: Not reached (1-year OS: 78%, 2-year OS: 71%)	ICI rechallenge: 75%	[33,34,35,36]
TNBC	Pembrolizumab (CPS ≥ 10): Increased ORR, MSI-H/MMR-deficient: 30.8%, ICI + Chemo: OR 1.35 (95% CI)	Pembrolizumab (MSI-H/MMR-deficient): 3.5 months, ICI + Chemo (ITT): HR 0.80 (95% CI), ICI + Chemo (PD-L1+): HR 0.70 (95% CI), ICI + Chemo (no prior CT): HR 0.53 (95% CI)	ICI + Chemo (ITT): HR 0.89 (95% CI), ICI + Chemo (PD-L1+): HR 0.80 (95% CI), ICI + Chemo (no prior CT): HR 0.81 (95% CI), PD-1/PD-L1 + Chemo: No significant OS improvement	Not specified	[37,38,39,40]

Summary of the overall response rate (ORR), progression-free survival (PFS), overall survival (OS), and disease control rate (DCR) for different cancer types treated with immune checkpoint inhibitors (ICIs). It highlights the therapeutic impact of ICI in NSCLC, HCC, melanoma, and TNBC, showing the differences in response rates and survival outcomes for each type of cancer.

**Table 3 cancers-17-01408-t003:** Key Strategies to Enhance Immune Checkpoint Inhibitor (ICI) Efficacy.

Strategy	Key Findings	References
Combination Therapy	Anti-PD-1 + anti-CTLA-4 a dual checkpoint blockade improves outcomes. CAR-T cell therapy and mRNA vaccines enhances immune responses. Oncolytic viruses combined with chemotherapy or radiotherapy target metastatic tumors	[61,62,63,64,65]
ICI + Chemotherapy	Platinum-based chemo + ICI improved PFS and OS in endometrial cancer, with dMMR patients benefiting most	[66]
Anti-Angiogenesis + ICIs	Atezolizumab + bevacizumab in NSCLC improved OS. Ramucirumab + pembrolizumab enhanced OS in NSCLC and gastric cancer (GC).	[67,68,69]
Biomarker-guided Therapy	PD-L1, TMB, MSI as predictors; High LDH levels linked to poor OS; HER2-negative GC had ORR 61.9%, DCR 96.8%, PFS of 9 months and OS of 27 months.	[70]
Oncolytic Viruses	In situ ADV/HSV-tk + SBRT + pembrolizumab in mNSCLC had ORR of 33.3%, CBR of 70.4%, PFS of 7.4 months and an OS of 18.1 months. H101 + nivolumab in HCC: ORR 11.1%, DCR 38.9%, PFS 2.69 months, OS 15.04 months.	[71,72]
Cancer Vaccines	mRNA-4157 + pembrolizumab in melanoma extended recurrence-free survival of 18 months. α-lactalbumin vaccine in TNBC induced immune responses with no major adverse events.	[73]
Radiotherapy + ICIs	Radioembolization + ICI in HCC had ORR of 89.5%, DCR of 94.7% and LDRT enhanced ICI responses in preclinical models.	[74]
JAK Inhibitors + ICIs	JAK inhibitors improved ICI response rate (53%) and 2-year PFS in Hodgkin lymphoma and NSCLC; JAK1 inhibition post-pembrolizumab enhanced immune function.	[75,76]
Microbiome Modulation	Gut microbiota regulates ICI response via metabolite production. Fecal microbiota transplant enhances PD-1 therapy. Bile acid metabolism increases CXCR6+ NKT cells in HCC.	[77,78,79]
Alternative Checkpoints (LAG3, TIM3, TIGIT)	LAG3 inhibitors (relatlimab, eftilagimod alpha and pembrolizumab) improved ORR 8.3%, DCR 33% Dual inhibition of LAG3 + TIGIT enhances CD8+ T-cell response and reduces tumor growth.	[80,81]
Metabolic Reprogramming + ICIs	Targeting glucose/lactate metabolism enhances ICI efficacy. Metformin + PD-1 inhibitor (nivolumab or pembrolizumab) improved NSCLC response rate.	[82,83,84,85,86,87,88]
Glutamine Metabolism + ICIs	High glutamine metabolism linked to ICI resistance. Hight GMScore in HCC correlated with poor OS and high immune checkpoint expression.	[89]

The table highlights key strategies in ICI therapy, including dual checkpoint blockade, chemotherapy, radiotherapy, metabolic modulation and biomarker-driven therapies. Various clinical trials and studies have demonstrated improvements in ORR, PFS, OS, and DCR when combined with additional treatments, such as anti-angiogenesis therapy, JAK inhibitors, microbiome modulation, and oncolytic viruses.

## Data Availability

Data are contained within this article.

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
