# Peer review of "Advancing Cancer Treatment: A Review of Immune Checkpoint Inhibitors and Combination Strategies"

_cancers, 2025, doi:10.3390/cancers17091408_

Round 1
Reviewer 1 Report
Comments and Suggestions for Authors
The Review by Dr Neil Valencia and Dr Seung Won Lee provides a comprehensive overview of immune checkpoints and possible combinatorial approaches in cancer therapy. The manuscript presents a wealth of data, but in some points the narrative seems fragmented. For instance, the transition from a general overview of ICI effectiveness to specific clinical studies (Namikawa et al., Zaemes et al., Lee et al.) feels abrupt. A smoother connection between sections could be beneficial by clearly explaining why these particular studies were selected and how they contribute to the overall thesis of the paragraph.
However, there are some points needing revisions:
- the text would benefit from editing for clarity, flow, and grammar, please correct the Sentence structure, the Use of abbreviations (some are not introduced clearly), Punctuation and run-on sentences, and Paragraph transitions.
- There is a comparison of different therapeutic strategies (BRAF/MEKi, anti-PD-1, PD-1/CTLA-4 combination), but the text lacks a critical reflection on costs, toxicity, and the patient profile that justifies the choice of one therapy over another. It would also be helpful to indicate the limitations of the studies cited, such as sample size, retrospective nature, or lack of randomization. About BRAF/MEKi, the authors should clarify why PFS is longer but OS is similar — this is scientifically interesting and merits brief interpretation. In addition, a recent manuscript (doi: 10.1038/s41586-022-04833-8) shows that Pharmacologic inhibition of AR signaling improved responses to BRAF/MEK-targeted therapy in male and female mice, whereas induction of AR signaling (via testosterone administration) was associated with significantly impaired response to BRAF/MEK-targeted therapy in males and females. Whether these data are obtained in in vivo models, together, these results have important implications for therapy. The authors have to add these important data.
- Sometimes the references are not well integrated into the text. Results are listed without providing a critical evaluation of the individual studies. For example, Zaemes et al. [31] and Lee et al. [32] are mentioned sequentially, but there is no comparative discussion or contextualization of the findings (e.g., patient heterogeneity, differences in TFS definition).
- The presented findings are intriguing but are not discussed in depth in terms of their practical implications. How can these data influence clinical decision-making? What is the impact on the therapeutic algorithm for patients with advanced melanoma?
- Whether authors take into considerations clinical results, they have to mention also important findings related to preclinical studies analyzing combinatorial approaches. In some types of cancer, for instance, there are some data concerning the combinatorial approaches of androgen antagonists and ICIs or a crosstalk between Immune-checkpoints and Androgen receptor. These studies provide evidence of sex-hormone mediated regulation of immune checkpoint molecules in vitro with potential ramifications for immunotherapies that should be mentioned, given the emerging number of data available: a) doi: 10.3389/fcell.2021.663130. AR activation impacts NF-kB signaling by increasing IkBα and preventing NF-kB translocation into the nucleus, reducing PD-L1 promoter activation; b) doi: 10.1038/s41419-025-07350-4. Melanoma cells depleted of the androgen receptor become more responsive to the most commonly used immunocheckpoint inhibitors /pembro, ipilimumab, atezolizumab, suggesting that the receptor dampens the immunotherapy efficacy; c) doi.org/10.1038/s41586-022-04522-6. Inhibition of AR activity in CD8 T cells prevented T cell exhaustion and improved responsiveness to PD-1 targeted therapy via increased IFNγ expression; d)https://doi.org/10.1136/esmoopen-2018-000344 On this basis, the authors should discuss new approaches of AR-based immune checkpoint blockade, which combine specific modulation of AR with immune checkpoint inhibitors.
- In general, the review lacks of discussion of data that the authors take into consideration.
I have detailed it.
Author Response
The Review by Dr Neil Valencia and Dr Seung Won Lee provides a comprehensive overview of immune checkpoints and possible combinatorial approaches in cancer therapy. The manuscript presents a wealth of data, but in some points the narrative seems fragmented.
- For instance, the transition from a general overview of ICI effectiveness to specific clinical studies (Namikawa et al., Zaemes et al., Lee et al.) feels abrupt. A smoother connection between sections could be beneficial by clearly explaining why these particular studies were selected and how they contribute to the overall thesis of the paragraph.
Response: Thank you for your insightful feedback. We have revised the manuscript (Section 2.3. Melanoma) to include a brief transition that clearly connects the general overview of ICI effectiveness to the specific clinical studies (Namikawa et al., Zaemes et al., and Lee et al.). This addition clarifies the rationale behind highlighting these studies and emphasizes their relevance in supporting our discussion of ICI performance across different cancer types.
However, there are some points needing revisions:
- the text would benefit from editing for clarity, flow, and grammar, please correct the Sentence structure, the Use of abbreviations (some are not introduced clearly), Punctuation and run-on sentences, and Paragraph transitions.
Response: Thank you for your valuable feedback. We have revised the manuscript to improve clarity, correct sentence structure, ensure proper introduction of abbreviations, and enhance punctuation and paragraph transitions.
- There is a comparison of different therapeutic strategies (BRAF/MEKi, anti-PD-1, PD-1/CTLA-4 combination), but the text lacks a critical reflection on costs, toxicity, and the patient profile that justifies the choice of one therapy over another. It would also be helpful to indicate the limitations of the studies cited, such as sample size, retrospective nature, or lack of randomization. About BRAF/MEKi, the authors should clarify why PFS is longer but OS is similar — this is scientifically interesting and merits brief interpretation.
Response: Thank you for the valuable suggestion. We have added a brief discussion and explain (Section 2.3. Melanoma; 2nd paragraph) why PFS is longer but similar OS observed when compared to combination immunotherapy.
- In addition, a recent manuscript (doi: 10.1038/s41586-022-04833-8) shows that Pharmacologic inhibition of AR signaling improved responses to BRAF/MEK-targeted therapy in male and female mice, whereas induction of AR signaling (via testosterone administration) was associated with significantly impaired response to BRAF/MEK-targeted therapy in males and females. Whether these data are obtained in in vivo models, together, these results have important implications for therapy. The authors have to add these important data.
Response: Thank you for your insightful comment. In response, we have briefly incorporated findings from the referenced manuscript (doi: 10.1038/s41586-022-04833-8) (Section 2.3. Melanoma; 3rd paragraph) highlighting that pharmacologic inhibition of AR signaling enhances responses to BRAF/MEK-targeted therapy in both sexes, whereas AR activation via testosterone impairs efficacy.
- Sometimes the references are not well integrated into the text. Results are listed without providing a critical evaluation of the individual studies. For example, Zaemes et al. [31] and Lee et al. [32] are mentioned sequentially, but there is no comparative discussion or contextualization of the findings (e.g., patient heterogeneity, differences in TFS definition).
Response: Thank you for the helpful comment. We have clarified the purpose of including the studies by Zaemes et al. and Lee et al. in the revised manuscript (Section 2.3 Melanoma). While the results may appear similar, these studies were included to underscore the consistent efficacy of ICI, particularly in melanoma, and to align these findings with previously reported outcomes such as PFS and OS observed in other cancer types.
- The presented findings are intriguing but are not discussed in depth in terms of their practical implications. How can these data influence clinical decision-making? What is the impact on the therapeutic algorithm for patients with advanced melanoma?
Response: Thank you for the suggestion. In response, we have provided a brief summary for each cancer type (Section 2 2. Role of Immune Checkpoint Inhibitors in Selected Cancers), highlighting the key findings and their potential clinical implications.
- Whether authors take into considerations clinical results, they have to mention also important findings related to preclinical studies analyzing combinatorial approaches. In some types of cancer, for instance, there are some data concerning the combinatorial approaches of androgen antagonists and ICIs or a crosstalk between Immune-checkpoints and Androgen receptor. These studies provide evidence of sex-hormone mediated regulation of immune checkpoint molecules in vitro with potential ramifications for immunotherapies that should be mentioned, given the emerging number of data available:
- a) doi: 10.3389/fcell.2021.663130. AR activation impacts NF-kB signaling by increasing IkBα and preventing NF-kB translocation into the nucleus, reducing PD-L1 promoter activation;
- b) doi: 10.1038/s41419-025-07350-4. Melanoma cells depleted of the androgen receptor become more responsive to the most commonly used immunocheckpoint inhibitors /pembro, ipilimumab, atezolizumab, suggesting that the receptor dampens the immunotherapy efficacy;
- c) doi.org/10.1038/s41586-022-04522-6. Inhibition of AR activity in CD8 T cells prevented T cell exhaustion and improved responsiveness to PD-1 targeted therapy via increased IFNγ expression;
d)https://doi.org/10.1136/esmoopen-2018-000344 On this basis, the authors should discuss new approaches of AR-based immune checkpoint blockade, which combine specific modulation of AR with immune checkpoint inhibitors.
Response: Thank you for your insightful comment. While we find the suggested preclinical studies on androgen receptor (AR) signaling and immune checkpoint regulation highly relevant and thought-provoking, we did not all incorporated except (doi: 10.1038/s41419-025-07350-4) in Section 2.3 Melanoma 1st paragraph. Findings in those publications are not primarily focus on cancer types the scope of our review. Nevertheless, we truly appreciate your recommendation and are deeply interested in this emerging area, particularly the potential of sex hormone-mediated regulation in immunotherapy. We will certainly consider exploring this in future work, possibly as the focus of a follow-up publication.
Reviewer 2 Report
Comments and Suggestions for Authors
This review aims to improve ICI-based therapies and guide future cancer research by identifying effective combination approaches and understanding the resistance mechanisms. The review’s relevance and usefulness are clear, as future oncology will likely focus on combined immunotherapy selectively inhibiting tumor growth while minimizing harm to healthy organs.
My comments and suggestions.
- The review is cumbersome and difficult to read. The information in sections 2-5 is largely repetitive and contains few new insights. These sections should be shortened or removed. All information from these sections could be included in the introduction in a condensed form.
- Section 6 subsections would benefit from tables, figures, and brief conclusions. A graphical abstract would be a helpful addition to this article.
- The title of subsection 6.2 “Biomarker Therapy” is misleading. Authors are talking here about biomarker-guided therapy, not the direct use of biomarkers in the treatment of disease.
Author Response
This review aims to improve ICI-based therapies and guide future cancer research by identifying effective combination approaches and understanding the resistance mechanisms. The review’s relevance and usefulness are clear, as future oncology will likely focus on combined immunotherapy selectively inhibiting tumor growth while minimizing harm to healthy organs.
My comments and suggestions.
- The review is cumbersome and difficult to read. The information in sections 2-5 is largely repetitive and contains few new insights. These sections should be shortened or removed. All information from these sections could be included in the introduction in a condensed form.
Response: Thank you for your constructive feedback. In response to your suggestion, Sections 2–5 have been significantly shortened and streamlined to improve readability and reduce redundancy. To maintain clarity while preserving essential content, we condensed key findings into a summary table highlighting the efficacy of ICIs across different studies.
- Section 6 subsections would benefit from tables, figures, and brief conclusions. A graphical abstract would be a helpful addition to this article.
Response: Thank you for your valuable feedback. In response, we have added brief concluding statements at the end of each subsection to enhance readability and reinforce key points. Additionally, we created figure (3.4. Cancer vaccines) and made table (3.7. Microbiome Modulation) to provide visual clarity and better support the discussed findings
- The title of subsection 6.2 “Biomarker Therapy” is misleading. Authors are talking here about biomarker-guided therapy, not the direct use of biomarkers in the treatment of disease.
Response: Thank you for your insightful observation. In response, we have revised the subsection title to “Biomarker-Guided Therapy” (3.2. Biomarker-guided Therapy) to more accurately reflect the content and align with your suggestion.
Round 2
Reviewer 1 Report
Comments and Suggestions for Authors
It Is improved